# A regularization term for slide correlation reduction in whole slide image analysis with deep learning

**Hongrun Zhang** [1]                                    HONGRUN.ZHANG@LIVERPOOL.AC.UK
**Yanda Meng** [1]                                         YANDA.MENG@LIVERPOOL.AC.UK
**Xuesheng Qian** [4]                                     XUESHENG.QIAN@INTELLICLOUD.AI
**Xiaoyun Yang** [5]                                       XYANG@REMARKHOLDINGS.COM
**Sarah E.Coupland**[*] [2,3]                              S.E.COUPLAND@LIVERPOOL.AC.UK
**Yalin Zheng**[*] [1]                                     YALIN.ZHENG@LIVERPOOL.AC.UK

[1] *Department of Eye and Vision Science, University of Liverpool, Liverpool, UK*

[2] *Liverpool Ocular Oncology Research Group, University of Liverpool, Liverpool, UK*

[3] *Liverpool Clinical Laboratories, Liverpool University Hospitals NHS Foundation Trust, Liverpool, UK*

[4] *Chinese Academy of Sciences (CAS) IntelliCloud Technology Co., Ltd., Shanghai, China*

[5] *Remark Holdings, London, UK*

## Abstract

To develop deep learning-based models for automatic analysis of histopathology whole slide images (WSIs), the atomic entities to be directly processed are often the smaller patches cropped from WSIs as it is not always possible to feed a whole WSI to a model given its enormous size. However, a trained model tends to relate the slide-specific characteristics to diagnosis results because a large number of patches cropped from the same WSI will share common slide features and thus have strong correlations between them, resulting in deteriorated generalization capability of the trained model. Current approaches to alleviate this issue include data pre-processing (stain normalization or color augmentation) and adversarial learning, both of which introduce extra complications in computations. Alternatively, we propose to reduce the impact of this issue by introducing a new regularization term to the standard loss function to reduce the correlation of the patches from the same WSI. It is intuitive and easy-to-implement and introduces comparably smaller computation overhead compared to existing approaches. Experimental results prove that the proposed regularization term is able to enhance the generalization capability of learning models and consequently to achieve better performance. The code is available in: https://github.com/hrzhang1123/SlideCorrelationReduction.

**Keywords:** deep learning, histopathology, whole slide image, over-fitting, uveal melanoma.

## 1. Introduction

There has been an increasing interest in using pathological whole slide images (WSIs) in the field of digital pathology (Litjens et al., 2016; Bandi et al., 2018; Zanjani et al., 2018). However, WSIs are characterized by their tremendously large sizes ranging from 10k×10k pixels to even 100k×100k pixels. Given this, when applying a deep learning-based algorithm for the automatic analysis of histopathology WSIs, the atomic entities to be directly processed are smaller cropped patches (Hou et al., 2016), and usually up to 1000s

---

[*] Corresponding authors

of patches can be obtained from just a single WSI. In contrast to the size, the number of WSIs available to train a learning-based model is often comparably small (e.g. < 50 in many cases). Meanwhile, patches from the same slide share significant features in terms of appearances or morphologies (Figure.1). The accumulative effort of these factors renders a significant issue that a learning-based model tends to link the diagnosis to the slide-specific features that are not diagnostically relevant. This situation will lead to severe over-fitting and undermine the model's capability of generalization, especially when the number of slides for training is small.

Three commonly-used approaches exist to relieve the negative effect of this issue: staining normalization (Magee et al., 2009; Macenko et al., 2009), color augmentation (Tellez et al., 2019), and slide domain adversarial (SDA) (Lafarge et al., 2017; Ganin et al., 2016). Both staining normalization and augmentation serve as pre-processing steps either to unify the staining appearances of patches from different slides, or to introduce varieties of color in patches as a data augmentation approach. These two methods require a comparably large amount of time to pre-process an image before feeding it into a model. SDA is implemented by adversarial training to extract the features of a patch that are agnostic to the slide where the patch is from. However, SDA requires extra network architectures and consequently needs more computational resources.

In this paper, we propose a new alternative approach to directly reduce the correlations of high-level representations of patches from the same slide. It is intuitive and can be implemented by just introducing a regularization term to standard loss functions during training. Compared with the above three approaches, it requires no extra cost in pre-processing of images and introduces no extra network modules. In addition, since there is only one hyper-parameter in the proposed regularization term to be fine-tuned, it is easier to search for the optimal configuration. In summary, the contributions of this paper are (1) a new regularization term to directly reduce correlations between patches from the same slide, in order to alleviate over-fitting and enhance generalization capability of learning models; and (2) proven performance demonstrated by an empirical validation on a WSI dataset of uveal melanoma.

In what follows, we refer to slide correlation reduction (SCR) to the method trained with the proposed regularization term.

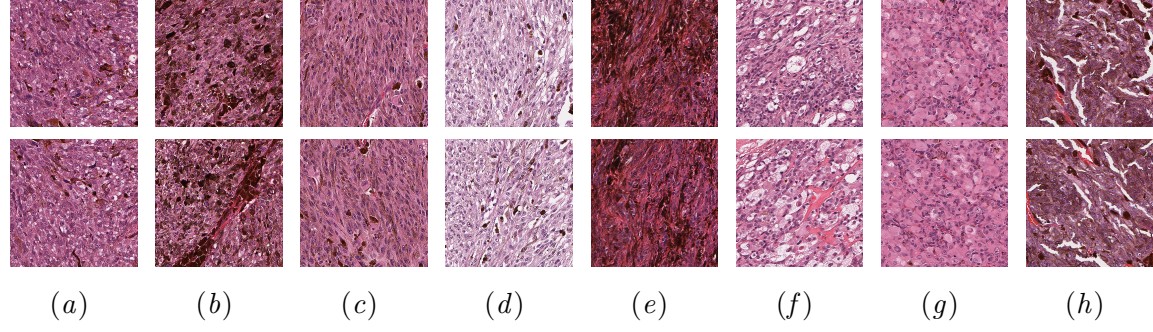

$(a)$ $\quad$ $(b)$ $\quad$ $(c)$ $\quad$ $(d)$ $\quad$ $(e)$ $\quad$ $(f)$ $\quad$ $(g)$ $\quad$ $(h)$

Figure 1: Eight pairs of patches from eight H&E slides of uveal melanoma.

## 2. Dataset and Method

### 2.1. Dataset Description

The proposed method is verified with a task to predict the nuclear *BAP1* (*nBAP1*) immuno-histochemical expression (positive or negative) of a section of a uveal melanoma (Zhang et al., 2020) on the basis of haematoxylin-and-eosin (H&E) stained slides only. Uveal melanoma (UM) is the most common primary intraocular malignancy in adults, and a high proportion of patients develop metastases to the liver, which is unfortunately incurable at present. Should the metastases be detected early, the UM patients can undergo liver surgery to prolong survival (Gomez et al., 2014; Marshall et al., 2013). A mutated *BAP1* gene is strongly associated with highly metastatic UM. Whilst this mutation can be determined using genetic analyses, immunohistology can be applied as a surrogate marker, whereby strong nuclear protein staining indicates that the *BAP1* gene is intact, and loss of nuclear staining is related to mutant *BAP1* (Kalirai et al., 2014; Farquhar et al., 2018).

This task is a specific case of a group of applications that aim to predict gene expression and mutations on the basis of histopathology slides using deep learning, which is an ongoing and booming field in recent years (Chen et al., 2020; Schmauch et al., 2020; Sun et al., 2019; Coudray et al., 2018), and they share similar problem contexts and corresponding methodologies. Gene mutations usually result in the same alterations in cellular morphology across the tissue region, which would imply that all the patches extracted from a WSI are usually assigned with the same label. In such a case, the over-fitting issue tends to be more server, given a large number of patches from the same slide that share similar features and have the same patch label.

In total, 184 cases of enucleated eyes were taken from pathology archives of the Royal Liverpool University Hospital, with each case including one tumour-representative slide being scanned at $40\times$ magnification. We randomly selected 140 slides (66 *BAP1* positive and 74 BAP1 negative) as the training set and 44 slides (16 *BAP1* positive and 28 *BAP1* negative cases) as the test set. For each slide, patches of $1024\times1024$ pixels were tiled from the tumor regions, and the *nBAP1* status of each patch was labeled by the corresponding slide *nBAP1* status. In total there were 99,778 patches for training and 30,693 patches for testing.

In the Appendix, we have provided the validation results on the Camelyon16 (Bejnordi et al., 2017) dataset with the task to detect lymph node metastases in women with breast cancer.

### 2.2. Correlations between patches from the same slide

As shown in Figure.1, patches from the same slide have high similarities in terms of appearance and morphology.

We conducted two experiments on the 140 slides for the empirical demonstration of highly correlations between patches cropped from the same slide. The patches cropped from the 140 slides were split into training set and validation set in two different ways. In the first experiment, we mixed the patches from different slides and randomly split them into training set and validation set; therefore patches from the same slide could exist in both training and validation set. Figure.2.(a) shows the performance of training and validation

over 10 epochs in terms of area under curve (AUC). For the second experiment, we split the dataset at the slide-level to avoid information leakage as of the first experiment. That is, all the patches from a slide were either in the training set or validation set, but would not co-exist in both sets. Figure.2.(b) illustrates the training and validation performances. We can see from Figure.2 that when splitting patches from the same slide into training set and validation set randomly, the performances on the validation set were synchronized with those on the training set over epochs and could achieve high AUC values (Figure.2.(a)). In contrast, when all patches from a slide exist only on one set, the performances on the validation set were significantly worse than that on the training set. These two figures imply the strong correlations of patches from the same slide and the shared features were learnt by the trained model as the diagnostic features.

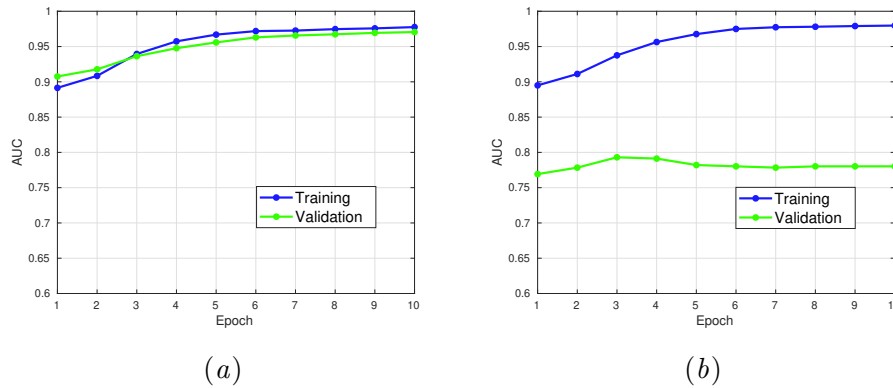

$(a)$ $(b)$

Figure 2: Area under curve values on training set and validation set over 10 training epochs. (a). Patches from different slides are randomly split into training set and validation set. (b). All the patches from a slide exist only either on training set or validation set. ResNet-18 is used.

### 2.3. The regularization term for slide correlation reduction

Consider $N$ patches extracted from different slides as a batch. The $i_{\text{th}}$ patch is denoted as $a_i$ $(i = 1, 2, ..., N)$ with its slide index $s_i$ that indicates the patch is from slide $s_i$. If patch $a_i$ and patch $a_j$ are from the same slide, then $s_i = s_j$. A vector feature $\boldsymbol{f}_i \in \mathbb{R}^{D \times 1}$ serves as the feature representation of patch $a_i$ extracted through a convolutional neural network, and $D$ is the dimension of the vector. The extracted features can be used for various downstream tasks. For the task considered specifically in this paper to predict the *nBAP1* status of uveal melanoma, the extracted features are fed into a classifier that generates prediction that indicates the probabilities of the corresponding patch to be *BAP1* positive and *BAP1* negative, denoted as $\boldsymbol{p}_i \in [0, 1]^2$. The loss function to train the network within the scope of the batch is formulated as,

$$\mathcal{L} = \frac{1}{N} \sum_i^N \mathcal{C}(\boldsymbol{p}_i, l_i), \tag{1}$$

where $l_i$ is the ground-truth label for patch $i$ and $\mathcal{C}$ is a criterion function that measures the distance of prediction $\boldsymbol{p}_i$ to the ground-truth label $l_i$. Cross-entropy is the most common criterion function for classification applications.

The contribution of this paper is a regularization term added to the $\mathcal{L}$, which quantifies the correlations of patches from the same slide,

$$\mathcal{L} = \frac{1}{N} \sum_i^N \mathcal{C}(\boldsymbol{p}_i, l_i) + \beta \mathcal{L}_{\mathrm{cr}}(\hat{\boldsymbol{F}}, \boldsymbol{s}) \tag{2}$$

where $\beta$ is a positive weight, $\hat{\boldsymbol{F}} \in \mathbb{R}^{D \times N}$ is the matrix that stacks the normalized feature vectors in the batch, i.e., the $i_{\mathrm{th}}$ column in $\hat{\boldsymbol{F}}$ is $\hat{\boldsymbol{f}}_i = \mathcal{N}(\boldsymbol{f}_i)$, with $\mathcal{N}$ being the operation that normalizes the values of the element in $\boldsymbol{f}_i$ to be between -1 and 1. And $\boldsymbol{s} \in \mathbb{R}^N$ is the vector that contains the slide indice information, i.e., the $i_{\mathrm{th}}$ element of $\boldsymbol{s}$ is $s_i$. The specific formation of the proposed regularization term is formulated as,

$$\begin{aligned}
\mathcal{L}_{\mathrm{cr}}(\hat{\boldsymbol{F}}, \boldsymbol{s}) &= \frac{1}{D} \sum_{0 < i < j \leq N} \hat{\boldsymbol{f}}_i^T \hat{\boldsymbol{f}}_j \, \mathrm{I}(i, j) \\
&= \frac{1}{D} \sum_{0 < i < j \leq N} \boldsymbol{u}_i^T \hat{\boldsymbol{F}}^T \hat{\boldsymbol{F}} \, \boldsymbol{u}_j \, \mathrm{I}(i, j)
\end{aligned} \tag{3}$$

where $\boldsymbol{u}_i \in \mathbb{R}^{N \times 1}$ is the one-hot vector with the $i_{\mathrm{th}}$ element being one and the rest all being zero. $T$ is the matrix transpose operation. $\mathrm{I}(i, j)$ is the indicator function,

$$\mathrm{I}(i, j) = \begin{cases} 1, & \text{if } s_i = s_j \quad (\text{i.e., from the same slide}) \\ 0, & \text{otherwise} \end{cases} \tag{4}$$

Equation (3) can be simplified into a matrix operation form as,

$$\mathcal{L}_{\mathrm{cr}}(\hat{\boldsymbol{F}}, \boldsymbol{s}) = \frac{1}{D} \, \mathcal{S}\big(\hat{\boldsymbol{F}}^T \hat{\boldsymbol{F}} \odot \boldsymbol{M}\big), \tag{5}$$

where $\mathcal{S}$ is the operation that sums up all the elements in a matrix, $\odot$ is the element-wise product, and $\boldsymbol{M} \in \mathbb{R}^{N \times N}$ is an upper-triangular matrix with the element of $i_{\mathrm{th}}$ row and $j_{\mathrm{th}}$ ($i < j$) column being defined as

$$M_{i,j} = \begin{cases} 1, & \text{if } s_i = s_j \\ 0, & \text{otherwise} \end{cases} \tag{6}$$

The $\hat{\boldsymbol{F}}^T \hat{\boldsymbol{F}}$ is exactly the Gramian matrix (Horn and Johnson, 2012), in which the value of the element of $i_{\mathrm{th}}$ row and $j_{\mathrm{th}}$ column being the correlation of $\hat{\boldsymbol{f}}_i$ and $\hat{\boldsymbol{f}}_j$, with a higher value suggesting the corresponding pair of features are more correlated. $\boldsymbol{M}$ serves to select the target correlation values in $\hat{\boldsymbol{F}}^T \hat{\boldsymbol{F}}$.

## 2.4. Interpretation

The two terms in Equation (2) work in an adversarial way in some sense. On the one hand, minimizing the first term results in searching for the subspaces of feature that are

diagnostically discriminative. However, the learnt subspaces inevitably incorporate the feature spaces related to specific slides that are not diagnostically relevant and consequently not informative to discriminate the category. Therefore it hampers a model being trained to be more generalized. The situation is even worse if the number of slides is particularly small. On the other hand, the proposed regularization term (the second term in Equation (2)) aims to drive the learnt features away from the subspace characterized by each individual slide. Note that the subspace of an individual slide feature also overlaps with diagnostic subspace, thus a proper weight $\beta$ is required to function the trade-off.

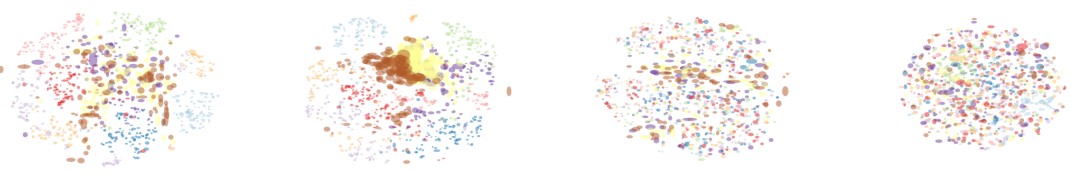

$(a)$ Without SCR. ResNet-18  $(b)$ Without SCR. ResNet-50  $(c)$ With SCR. ResNet-18  $(d)$ With SCR. Resnet-50

Figure 3: t-SNE distribution of the learnt features of patches from 10 slides, with and without slide correlation reduction (SCR), respectively. For each slide 100 patches are considered, and the same color refers to patches from the same slide. Features closed to each other in the 2-dimensional space are merged to a larger ellipsoid.

## 3. Experiments

### 3.1. Configurations

Five deep learning architectures were utilized as the backbone feature extractors in the experiments, namely ResNet-18, ResNet-50 (He et al., 2016), DenseNet-121 (Huang et al., 2017), AlexNet (Krizhevsky et al., 2012) and VGG-16 (Simonyan and Zisserman, 2014), and all were pre-trained with ImageNet dataset (Deng et al., 2009).

Patches were split into training set and test set based on slides. All the patches were resized to 256×256 pixels, and random rotation was adopted as the data augmentation approach during the training process. Each model was trained for 10 epochs with an initial learning rate of $5e-4$ and then $1e-4$ after epoch 5. Stochastic gradient descent (SGD) was used as the optimizer with a weight decay of $1e-4$. The slide-level performances are reported which were obtained from the mean values of predictions of all patches in a slide. Each performance value was the mean value of the results of 5 independent experiments. The $\beta$ was set to be 0.2. For all the experiments 0.5 was adopted as the threshold to compute the performance metrics except for AUC.

### 3.2. Performance

Table.1 presents the results of the proposed method in comparison of the baseline (no SDA nor SCR applied), SDA (Lafarge et al., 2017) and the proposed SCR by using different

backbone architectures as the feature extractors. For the most informative performance metric AUC, the proposed method is significantly superior to the baseline and SDA with all the backbone architectures, and with large margins in most cases and can up to 8% (ResNet-50). For all the other performance metrics, the proposed method is the best or closed to the best.

In Table.2, stain normalization (SN) and color jitter (CJ) were utilized as the additive pre-processing approaches to the baseline method, SDA, and the proposed SCR, respectively. CJ served in a way as data augmentation and was implemented by multiplying the values of brightness, contrast, saturation, and hue of an image with a random coefficient. The random coefficient was drawn each time between 0.8 and 1.2 with uniform probability. For the baseline method, the results show the CJ works better than SN. When combined CJ with the proposed SCR, the performance can further be improved, since except for the baseline method using DenseNet-121 (0.955 vs 0.947), the proposed method dominates the other two in AUC. In particular, when using VGG-16 as the backbone, the proposed method with CJ achieves the highest AUC values of all (0.968).

| Network | Method | Accuracy | Recall | Specificity | F1 | AUC |
|---------|--------|----------|--------|-------------|-----|-----|
| Resnet18 | Baseline | $0.649_{0.011}$ | $0.875_{0.001}$ | $0.521_{0.017}$ | $0.645_{0.007}$ | $0.776_{0.015}$ |
| | SDA | $0.600_{0.027}$ | $0.812_{0.001}$ | $0.478_{0.042}$ | $0.596_{0.016}$ | $0.772_{0.015}$ |
| | SCR | $0.622_{0.031}$ | $0.887_{0.024}$ | $0.471_{0.057}$ | $0.631_{0.017}$ | $\mathbf{0.834}_{0.002}$ |
| Resnet50 | Baseline | $0.519_{0.007}$ | $0.955_{0.028}$ | $0.270_{0.026}$ | $0.591_{0.005}$ | $0.801_{0.021}$ |
| | SDA | $0.545_{0.038}$ | $0.574_{0.025}$ | $0.528_{0.052}$ | $0.479_{0.028}$ | $0.650_{0.018}$ |
| | SCR | $0.584_{0.031}$ | $0.937_{0.001}$ | $0.382_{0.049}$ | $0.621_{0.018}$ | $\mathbf{0.889}_{0.003}$ |
| AlexNet | Baseline | $0.813_{0.029}$ | $0.887_{0.053}$ | $0.771_{0.069}$ | $0.776_{0.021}$ | $0.889_{0.004}$ |
| | SDA | $0.895_{0.018}$ | $0.899_{0.030}$ | $0.892_{0.039}$ | $0.862_{0.018}$ | $0.916_{0.001}$ |
| | SCR | $0.795_{0.062}$ | $0.912_{0.050}$ | $0.728_{0.120}$ | $0.769_{0.049}$ | $\mathbf{0.920}_{0.006}$ |
| DenseNet121 | Baseline | $0.836_{0.009}$ | $0.737_{0.025}$ | $0.892_{0.001}$ | $0.766_{0.016}$ | $0.880_{0.005}$ |
| | SDA | $0.859_{0.026}$ | $0.812_{0.068}$ | $0.885_{0.014}$ | $0.806_{0.043}$ | $0.881_{0.003}$ |
| | SCR | $0.822_{0.017}$ | $0.887_{0.025}$ | $0.785_{0.039}$ | $0.784_{0.012}$ | $\mathbf{0.918}_{0.005}$ |
| VGG16 | Baseline | $0.695_{0.027}$ | $0.875_{0.001}$ | $0.592_{0.042}$ | $0.676_{0.019}$ | $0.891_{0.007}$ |
| | SDA | $0.672_{0.030}$ | $0.75_{0.001}$ | $0.628_{0.048}$ | $0.625_{0.021}$ | $0.809_{0.002}$ |
| | SCR | $0.695_{0.018}$ | $0.937_{0.001}$ | $0.557_{0.028}$ | $0.691_{0.012}$ | $\mathbf{0.893}_{0.010}$ |

Table 1: Performance of the baseline method, slide domain adversarial (SDA) and the proposed regularization term of slide correlation reduction (SCR). The subscripts are the standard deviation values. The best AUC values are in bold.

### 3.3. Feature distribution

Figure.2.4 presents the feature distributions obtained by mapping the high-dimensional features to 2-dimensional using t-SNE (Van der Maaten and Hinton, 2008). As can be seen, when without using SCR for training, the learnt features of the patches from the same slide have smaller inter-distances in the feature space, and tend to cluster with each other. Such clustering is more significant with larger networks such as ResNet-50 since it has higher learning capability. This phenomenon suggests the slide-specific features are inevitably encoded in the learnt presentations of the patches. In contrast, with the proposed SCR, the

learnt features of the patches from the same slide distribute more evenly over the feature space, and present weaker spatial clues to infer they are from the same slide. It implies the slide-specific features among the same slide patches have been deprived from the learnt features to some extent.

| Network | Method | Accuracy | Recall | Specificity | F1 | AUC |
|---------|--------|----------|--------|-------------|-----|-----|
| Resnet18 | Baseline+SN | $0.850_{0.011}$ | $0.774_{0.030}$ | $0.892_{0.001}$ | $0.789_{0.018}$ | $0.890_{0.002}$ |
| | Baseline+CJ | $0.873_{0.016}$ | $0.892_{0.028}$ | $0.862_{0.012}$ | $0.8367_{0.021}$ | $0.927_{0.002}$ |
| | SDA+CJ | $0.899_{0.011}$ | $0.937_{0.001}$ | $0.878_{0.017}$ | $0.872_{0.012}$ | $0.9334_{0.002}$ |
| | SCR+CJ | $0.889_{0.022}$ | $0.928_{0.021}$ | $0.867_{0.031}$ | $0.859_{0.026}$ | $\mathbf{0.951}_{0.005}$ |
| Resnet50 | Baseline+SN | $0.854_{0.023}$ | $0.800_{0.025}$ | $0.885_{0.026}$ | $0.800_{0.029}$ | $0.899_{0.006}$ |
| | Baseline+CJ | $0.777_{0.009}$ | $0.937_{0.002}$ | $0.685_{0.014}$ | $0.753_{0.007}$ | $0.936_{0.005}$ |
| | SDA+CJ | $0.809_{0.018}$ | $0.862_{0.025}$ | $0.778_{0.034}$ | $0.766_{0.016}$ | $0.919_{0.008}$ |
| | SCR+CJ | $0.845_{0.017}$ | $0.937_{0.001}$ | $0.792_{0.026}$ | $0.815_{0.016}$ | $\mathbf{0.953}_{0.002}$ |
| AlexNet | Baseline+SN | $0.831_{0.023}$ | $0.800_{0.072}$ | $0.850_{0.014}$ | $0.774_{0.040}$ | $0.887_{0.004}$ |
| | Baseline+CJ | $0.850_{0.011}$ | $0.812_{0.001}$ | $0.871_{0.017}$ | $0.797_{0.012}$ | $0.911_{0.003}$ |
| | SDA+CJ | $0.799_{0.030}$ | $0.875_{0.001}$ | $0.757_{0.047}$ | $0.761_{0.027}$ | $0.915_{0.003}$ |
| | SCR+CJ | $0.859_{0.022}$ | $0.875_{0.039}$ | $0.850_{0.052}$ | $0.819_{0.019}$ | $\mathbf{0.932}_{0.002}$ |
| DenseNet121 | Baseline+SN | $0.768_{0.009}$ | $0.600_{0.030}$ | $0.864_{0.014}$ | $0.652_{0.018}$ | $0.877_{0.002}$ |
| | Baseline+CJ | $0.836_{0.009}$ | $0.862_{0.025}$ | $0.821_{0.001}$ | $0.792_{0.014}$ | $\mathbf{0.955}_{0.003}$ |
| | SDA+CJ | $0.836_{0.009}$ | $0.875_{0.001}$ | $0.814_{0.014}$ | $0.795_{0.008}$ | $0.929_{0.002}$ |
| | SCR+CJ | $0.863_{0.014}$ | $0.850_{0.030}$ | $0.871_{0.017}$ | $0.819_{0.019}$ | $0.947_{0.004}$ |
| VGG16 | Baseline+SN | $0.795_{0.020}$ | $0.612_{0.061}$ | $0.899_{0.014}$ | $0.683_{0.041}$ | $0.870_{0.007}$ |
| | Baseline+CJ | $0.745_{0.037}$ | $0.762_{0.027}$ | $0.735_{0.059}$ | $0.685_{0.031}$ | $0.863_{0.010}$ |
| | SDA+CJ | $0.777_{0.033}$ | $0.8125_{0.055}$ | $0.757_{0.057}$ | $0.726_{0.034}$ | $0.874_{0.017}$ |
| | SCR+CJ | $0.872_{0.011}$ | $0.837_{0.030}$ | $0.892_{0.001}$ | $0.826_{0.017}$ | $\mathbf{0.968}_{0.003}$ |

Table 2: Performance of the baseline method, slide domain adversarial (SDA) and the proposed slide correlation reduction (SCR), with stain normalization (SN) and color jitter (CJ) serving as the extra pre-precossing methods. The subscripts are the standard deviation values. The best AUC values are in bold.

## 4. Conclusion

In this paper, we propose an intuitive and easy-to-implement regularization term to be added to the standard loss function, in order to reduce the correlation of patches from the same slide, and in turn to increase the generalization capability of deep learning models. We have applied this new approach for the analysis of histopathology WSIs for the prediction of $nBAP1$ status. Indeed, it offers improved performance compared to existing approaches. It is compatible and effective for a variety of existing network architectures. This SCR is expected to be extendable to wider applications when the correlation is of concern.

## Acknowledgments

We thank a large number of people associated with this project, including the uveal melanoma patients. Zhang H. thanks Chinese Academy of Sciences IntelliCloud Technology Co., Ltd. for the industry studentship.

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

## Appendix A. Ablation Study

We selected the cases with SCR and with color jitter as the pre-processing, which achieved the best performance of all (see Table.2 in the main paper), to explore how performances vary with different values of $\beta$ in Eq.(2). Figure.(4) shows the performances have peaks around $\beta = 0.2$ and slightly decrease with the increase of $\beta$. However, for a wide range values of $\beta$ better performances can be achieved than the one without SCR regularization term ($\beta = 0$).

To further demonstrate it is exactly the reduction in slide correlation functions that improves the generalization capability of a deep learning model, we conducted experiments that instead of reducing slide correlations, enhanced slide correlations, simply by reversing the plus sign to minus sign in Equation.(2). Figure.(3) presents the corresponding AUC values, which shows by enhancing slide correlations (denoted as SCE) the performances are significantly worse than by reducing slide correlations, and in some cases, it even has lower AUC values than the baseline.

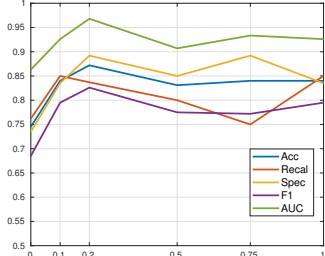

| Method | Baseline | SCE | SCR |
|---|---|---|---|
| ResNet18 | 0.776 | 0.788 | 0.834 |
| ResNet50 | 0.801 | 0.803 | 0.889 |
| AlexNet | 0.889 | 0.852 | 0.92 |
| DenseNet121 | 0.88 | 0.885 | 0.918 |
| VGG16 | 0.891 | 0.841 | 0.893 |

Figure 4: Performances of slide correlation reduction using VGG16 with different values of $\beta$.

Table 3: AUC values of the baseline, slide correlation enhance (SCE) and slide correlation reduction (SCR) methods, respectively.

## Appendix B. Comparisons of the AUC values on training set and test set

Table.4 presents the AUC values on the training set and test set, respectively. Without using SCR, the baseline method can achieve extremely high values of AUC on the training set, which are significantly better than those with SCR. In contrast, the performances on the test set of the baseline method are inferior to those with SCR. It demonstrates that when trained with the proposed SCR, the issue of over-fitting can be alleviated to some extent, and consequently the trained model is able to obtain higher generalization capability.

## Appendix C. Validation on CAMELYON16 dataset

The Camelyon16 dataset (Bejnordi et al., 2017) contains 270 WSIs (160 normal and 110 tumor) for training, and 130 WSIs for testing (81 normal and 49 tumor). We followed (Li and Ping, 2018) to use the first 140 normal slides and the first 100 tumor slides for training, and the remaining slides for validation. 50,000 patches were extracted from the normal

| Method | without SCR | | with SCR | |
|---|---|---|---|---|
| | Training | Test | Training | Test |
| ResNet18 | 0.981 | 0.927 | 0.956 | 0.951 |
| ResNet50 | 0.995 | 0.936 | 0.981 | 0.953 |
| AlexNet | 0.962 | 0.911 | 0.937 | 0.932 |
| DenseNet121 | 0.993 | 0.955 | 0.977 | 0.947 |
| VGG16 | 0.992 | 0.863 | 0.986 | 0.968 |

Table 4: AUC values on the training set and test set with and without the slide correlation reduction (SCR) method. Both are with color jitter for pre-processing.

and tumor slides in the training set, respectively (In total 100000 patches for training). From the validation set, 10000 normal patches and 10,000 tumor patches were extracted for validation. All the patches were from the 40X magnification and with the size of 256 x 256 pixels. Random cropping to 224 x 224 pixels and random rotation/flipping were utilized as the data augmentations. The networks were trained for 15 epochs with a constant learning rate of 0.001. For more details please refer to the released code. Table.5 presents the patch-level AUC values on the validation set. We can observe that the proposed SCR achieves higher AUC values than the baseline method and the slide domain adversary (SDA) using the two backbone networks (Resnet-18 and Resnet-50) with and without color jitter (CJ) as the data augmentation.

| | Resnet18 | Resnet18 (CJ) | Resnet50 | Resnet50 (CJ) |
|---|---|---|---|---|
| Baseline | $0.910_{0.002}$ | $0.922_{0.003}$ | $0.909_{0.005}$ | $0.926_{0.003}$ |
| SDA | $0.906_{0.006}$ | $0.923_{0.002}$ | $0.918_{0.003}$ | $0.929_{0.003}$ |
| SCR | $0.923_{0.002}$ | $0.934_{0.001}$ | $0.922_{0.004}$ | $0.931_{0.002}$ |

Table 5: Patch-level AUC values of the baseline method, SDA and the proposed SCR on the validation set of Camelyon16, respectively. CJ: color jitter.

