# OpenReview forum: "A regularization term for slide correlation reduction in whole slide image analysis with deep learning"
_MIDL.io/2021/Conference — MIDL 2021_

### Official Review · AnonReviewer3 · 2021-03-06

**Confidence:** 5
**Preliminary Rating:** 1

**Summary:**

In the paper "A regularization term for slide correlation reduction in whole slide image analysis with deep learning" the authors focused on the investigation of the impact of tiles correlation on the whole slide analysis task. The proposed own way to improve this aspect of the whole slide classification. The study used 140 slides and compared five deep learning architectures as a backbone.

**Strengths:**

The aspect of regularization and high correlation between patches has an important role during training deep learning models for whole slide image analysis. This research area is not well study, and progress in this matter is needed.  The authors focused on a task that can be interesting for readers.

**Weaknesses:**

The paper is confusing and not precise in many places. The method is not well described, and by this, it is difficult to replicate the experiments. The proposed method was evaluated using a single database, which is the main weakness.  The method, that gives to be a general solution for a specific task, should be evaluated on various, independent datasets in order to present method robustness. The experiments are not well designed and do not provide answers to formulate research questions.  Section 2.2 presented the impact of various patch selection strategies on training performance. It is a well-known fact that using patches from the same distribution, in both training and validation sets, we will get higher training performance (because in both sets we have similar data). This section is not present the impact of correlation between patches, as suggested at the beginning of the section. The similarity/ correlation between patches from the same slide can be investigated using other metrics, for example, FID.

Example of confusing parts:
1. " Meanwhile, patches from the same slide share significant features in terms of appearances or morphologies"- that sentence is only partially correct. Single slide, for example, a prostate biopsy can include tissue from other organs (such as the colon), or many specimens can include fat are that is not similar to the tissue area. So, the whole area of tissue from a single slide is not similar. Some patches/ part of patches are very similar, but not all of them.
2. "Three commonly-used approaches exist to relieve the negative effect of this issue: staining normalization "--> the stain normalization is not solving the over-fitting issue, but it can increase robustness.
3. "Both staining normalization and augmentation serve as pre-processing steps either to unify
the staining appearances of patches from different slides, or to introduce varieties of color in
patches as a data augmentation approach. These two methods require a comparably large
amount of time to pre-process an image before feeding it into a model"--> These two operations are applied in various ways, on a different method stage. The augmentation is applied only during method development and can increase the time of trainng. Moreover, typically from one patch (tile) after augmentation, we got many patches. The normalization is typically applied when the method is used on the new dataset ( the test dataset), and then each patch of the new dataset is normalized using a dedicated method. There are several methods to normalize data, and they have various complexity.
4. the title of the paper is confusing, in the paper described method focused on patch correlation, however, the title mentioned slide correlation. It is not clear why.
5. The goal of the paper is the reduction of correlation between patches, but in section 3.1, the authors mentioned an application of augmentation. It is not clear how used augmentation was monitored in terms of correlation.



**Deanonymize Review:**

no

**Detailed Comments:**

Not all citations are used appropriately, for example: "There has been an increasing interest in using pathological whole slide images (WSIs) in the field of digital pathology (Bandi et al., 2018)"- the paper  (Bandi et al., 2018)is about the detection of metastases and lymph node classification, not about trends in digital pathology. There are several other papers, that take into account this aspect.

**Justification Of The Preliminary Rating:**

The paper is confusing and not precise in many places. The method is not well described, and by this, it is difficult to replicate the experiments. The proposed method was evaluated using a single database, which is the main weakness.  The method, that gives to be a general solution for a specific task, should be evaluated on various, independent datasets in order to present method robustness. The experiments are not well designed and do not provide answers to formulate research questions.

**Paper Type:**

methodological development

**Special Issue:**

no

---

> ### Author Response · Authors · 2021-03-17
> **Response to Reviewer 3, Part 1 (1/2).**
>
>
> &nbsp;
>
> Thank you for reviewing the paper.
>
> &nbsp;
>
> * In weakness: “The proposed method was evaluated using a single database, which is the main weakness.”
>
> **Author Response #1**: Please see our comments to Reviewer #4.
>
> This paper can be seen as a proof-of-principle and our work verified that the proposed idea works in the uveal melanoma dataset. However, we agree that it is a good idea to verify the proposed on wider tasks, to generalize the proposed method, and plan this in our future work.
>
> &nbsp;
>
> * In weakness: “Section 2.2 presented the impact of various patch selection strategies on training performance. It is a well-known fact that using patches from the same distribution, in both training and validation sets, we will get higher training performance.  This section is not present the impact of correlation between patches, as suggested at the beginning of the section. The similarity/ correlation between patches from the same slide can be investigated using other metrics, for example, FID”
>
> **Author Response #2**:  We did not aim to quantitatively evaluate the correlations in Section2.2. The experiment presented is an empirical demonstration to support the reasoning presented in the introduction, i.e. that patches from same slide have highly correlation. From the information perspective, the conditional entropy H(x|y) tends to be very small, when x, y are two patches from same slide. There exist shared features among the patches from the same slides that are not diagnostically relevant. However, since the number of patches from the same slide is typically large, these features are incorrectly learnt by the model as the diagnostic features. That is why the validation performances in Fig.2(a) are almost the same with the training performance. However, the shared features hamper a learnt model to be more generalized to unseen  slides, and the proposed method aims to alleviate the negative impacts of these features.
>
> &nbsp;
>
> * In weakness: " ‘Meanwhile, patches from the same slide share significant features in terms of appearances or morphologies’- that sentence is only partially correct. "
>
> **Author Response #3**: We agree with you that not all patches in a slide are visually similar. However: (1) there can be some similarities amongst the patches from the same slide, which are not significantly visible (eg. noise in a certain range of frequency spectrum), maybe resulted from the slide producing and scanning process. This argument is qualitatively supported by the t-sne figures with no SCR shown in the paper, because most of the learnt features of the 100 randomly selected patches from the same slides are clustering very closed to each other, indicating the highly similarity amongst them. (2). If the patches from a slide are not similar to each other at all, the issue of slide domain generalization will not exist. In reality, it is highly likely that there are large local regions in a slide cropped from which the patches are similar. (3) It is not necessary to require all patches in a slide to be similar for the proposed regularization term to be effective. For those less similar patches from a slide, intuitively they will not have highly correlated high-level representation; therefore, their pairs will not have contributed large values to the proposed regularization term.
>
> &nbsp;
>
> * In weakness: “the stain normalization is not solving the over-fitting issue, but it can increase robustness.”
>
> **Author Response #4**: We have learnt from our experiences that stain normalization can relieve the issue of over-fitting.  Suppose without staining normalization operation on training set and test set, a model is likely to be trained to be over-fitting to slide-specific features that are not diagnostically relevant, i.e., the model will recognize these as the diagnostic features. This situation is even worse if the number of slides is small. When inferred on the test set, the trained model will tend to search for the similarity with these learnt features from the samples in the test set and accordingly make the predictions. This is a typical over-fitting case, and is the main topic this paper is talking about. Staining normalization can work to minimize the differences in colors between the samples in training set and test set, and consequently relieve the over-fitting issue.
>
> Finally, with respect to “robustness” which you mentioned: could you please explain more what does robustness specifically refer to?

---

> ### Author Response · Authors · 2021-03-17
> **Response to Reviewer 3, Part 2 (2/2).**
>
> &nbsp;
>
> * In weakness: " ‘Both staining normalization and augmentation serve as pre-processing steps either to unify the staining appearances of patches from different slides, or to introduce varieties of color in patches as a data augmentation approach. These two methods require a comparably large amount of time to pre-process an image before feeding it into a model’--> These two operations are applied in various ways, on a different method stage. The augmentation is applied only during method development and can increase the times of trainning. Moreover, typically from one patch (tile) after augmentation, we got many patches. The normalization is typically applied when the method is used on the new dataset ( the test dataset), and then each patch of the new dataset is normalized using a dedicated method. There are several methods to normalize data, and they have various complexity.”
>
> **Author Response #5**: Apologies but there may appear to be some misinterpretation. What you describe are the specific steps to do the staining normalization and augmentation, and they are exactly the pre-processing steps, i.e., the operations on images before feeding a model. What are the conflicts with the two sentences you cited from the paper? Could you please explain more?
>
> Further, it is better to apply stain normalization on both training set and test set, not only just test set: see [Ciomni et al].
>
> ###### Ciompi, F., Geessink, O., Bejnordi, B.E., De Souza, G.S., Baidoshvili, A., Litjens, G., Van Ginneken, B., Nagtegaal, I. and Van Der Laak, J., 2017, April. The importance of stain normalization in colorectal tissue classification with convolutional networks. In 2017 IEEE 14th International Symposium on Biomedical Imaging (ISBI 2017) (pp. 160-163). IEEE.
>
> &nbsp;
>
> * In weakness: 'the title of the paper is confusing, in the paper described method focused on patch correlation, however, the title mentioned slide correlation. It is not clear why.'
>
> **Author Response #6**: ‘slide correlation reduction’ refers to reducing the correlations of patches from the same slide. The shared features among these patches are characterized by the slide.
>
> &nbsp;
>
> * In weakness: “The goal of the paper is the reduction of correlation between patches, but in section 3.1, the authors mentioned an application of augmentation. It is not clear how used augmentation was monitored in terms of correlation.”
>
> **Author Response #7**: Reduction of correlation between patches from the same slide serves as an approach to drive the learnt feature away from the slide-specific feature space. The data augmentation mentioned in section 3.1, the random rotation, cannot work for this purpose.
>
> &nbsp;
>
> * In detailed comments: “Not all citations are used appropriately, for example: "There has been an increasing interest in using pathological whole slide images (WSIs) in the field of digital pathology (Bandi et al., 2018)"
>
> **Author Response #7**: Apologies if this was not clear: the paper cited (Bandi et al) is an example of specific application of WSIs in digital pathology. It is in accordance with the topic “an increasing interest in using pathological whole slide images”. However, we will also add more related papers for this aspect in the revision. Thank you for your suggestion.
>
> If there are other ore pertinent examples, we would welcome the reviewer to specifically point these out, so that we can improve the quality of this paper.

---

### Official Review · AnonReviewer4 · 2021-03-06

**Confidence:** 5
**Preliminary Rating:** 4
**Recommendation:** Oral
**Final Rating:** 4

**Summary:**

The paper addresses the problem of domain generalisation in histopathology images where every slide is considered a separate domain. This is a very relevant problem in histopathology. The paper presents a regularisation method that reduces the correlation of the learned features within a single batch. The proposed method is well outlined and motivated and the results are convincing.

**Strengths:**

The presented method is a relatively straightforward regularisation method that is well motivated and seems to successfully address the slide-domain issue when training with patches from histopathology images. The authors present an extensive number of experiments and comparison to a baseline and two other methods.

**Weaknesses:**

I do not have any major weakness to report. One nice addition would be to add additional classification tasks. Perhaps some that are well established and ideally using public datasets. That will make the results more convincing.

**Deanonymize Review:**

no

**Detailed Comments:**

Minor issue:
- Instead p_i \in R^2 it should be p_i \in [0, 1] as p_i is a probability.

**Final Rating Justification:**

While the authors did not implement the suggested experiments, I still find the work valuable and of interest for the community.

**Justification Of The Preliminary Rating:**

I find this very intuitive method and valuable addition to the set of methods that can be used to address the domain generalisation issue in histopathology. While more experiments for different tasks would be more convincing, the proposed method is of interest to the community.

**Paper Type:**

methodological development

**Questions To Address In The Rebuttal:**

The authors should make a comparison for the same tasks and datasets as in Lafarge et al.

**Special Issue:**

no

---

> ### Author Response · Authors · 2021-03-17
> **Response to Reviewer 4**
>
> &nbsp;
>
> The authors feel very inspired when the work is appreciated. Thank you for appreciating the novel idea of this paper and recognizing its merits, and also for providing the constructive suggestions.
> &nbsp;
>
> * In weakness: “One nice addition would be to add additional classification tasks. Perhaps some that are well established and ideally using public datasets. That will make the results more convincing”
>
> **Author Response #1**:  Our paper can be seen as a proof-of-principle project, whereby we demonstrate its effectiveness addressing the uveal melanoma classification problem. However, we agree with you that it will make the proposed method more transposable, if the proposed method is verified in broader tasks. Therefore, we aim to extend our work further on other tasks.
>
> On the other hand, this comment has highlighted a fundamental issue in medical image applications – i.e. ethical approval. It’s our plan to share the data for future studies with appropriate ethical approval.
>
> &nbsp;
>
> * “The authors should make a comparison for the same tasks and datasets as in Lafarge et al.”
>
> **Author Response #2**: Thank you for this suggestion. We are considering other tasks for further verifying the proposed method, and believe it is a good idea to compare with SDA [Lafarge et al] in the mitoses task, and include this in the revision.

---

### Official Review · ~Christopher_P_Bridge1 · 2021-03-07

**Confidence:** 4
**Preliminary Rating:** 3
**Recommendation:** Oral
**Final Rating:** 3

**Summary:**

This paper introduces a new regularisation term designed for improving the generalisation performance of neural network models trained on highly correlated data, such as commonly found in histopathology whole slide images. The regularisation term directly penalizes the correlation between the learned features of input samples drawn the same slide. Experiments on uveal melanoma images and multiple deep learning architectures demonstrate that in many cases the models trained using the regularisation term outperform other methods.

**Strengths:**

The proposed method is very elegant and simple to apply. The message of the paper is very concisely communicated. There are appropriate experiments demonstrating the advantages of the method over other approaches.


**Weaknesses:**

A key weakness of the paper is that although it is primarily a methods paper, the method is only validated on a single, highly-specialised use case using a private dataset. This means that it is difficult to know the extent to which the results will generalise to other potential use cases of interest.

There are several missed opportunities for experiments that would provide more insight into the method (see below).


**Deanonymize Review:**

yes

**Detailed Comments:**

"Criterion function" is highly unusual terminology - "loss function" should be preferred.

Although generally speaking the explanation of the method is clear there are a couple of important omissions. Firstly it is never stated from where in the models the feature vectors that the SCR regularisation are applied to are drawn from. Is it the final layer before the output layer in all cases? Would there be advantages to applying to the output of multiple layers in the model? Was this investigated?

Another key omission is that the batch size parameter used during training is not given. This is an especially important parameter for this model since the SCR is only applied between patches in the same batch. Assuming that the elements are randomly and independently chosen, if the batch size is considerably smaller than the number of training cases a large proportion of batches will have no patches drawn from the same slide, all elements of the M matrix will be 0 and the regularisation term will have no effect. It would be very interesting to see, therefore, what the effect of batch size on the effectiveness of the SCR would be, and whether deliberately creating the batches to ensure that there are large number of patches that originate from the same slide would increase the size of the observed effect.

It would also be interesting to know how the regularisation term affects the performance on the training set, and also on the stability of the training process, since as the authors state to a certain extent the main loss function and the SCR term are opposed to each other. Would it be possible to provide training curves like figure 2 for models using SCR? Perhaps one could be added as panel 2c.

It is rather confusing that the AUC scores in tables 1 and 2 do not seem to be highly correlated with the accuracy values. Is this because the resulting models are not well calibrated for some reason? Do the authors have any insight into this they can share?





**Final Rating Justification:**

My original review still stands. This is a nice methodology that is mostly clearly explained, but it is let down by limited experiments. I see that some other reviewers found the methodology confusing. I agree that there are some things that they have highlighted that would improve the clarity, but these are easily addressable. Overall I found the explanation of the method clear.

**Justification Of The Preliminary Rating:**

An elegant method that may be of broad utility for those working on pathology images and other applications with highly correlated training data. Solid experimental evaluation demonstrating improvement over state of the art methods.



**Paper Type:**

methodological development

**Questions To Address In The Rebuttal:**

Do the authors intend to make an implementation of their regularisation term available?


**Special Issue:**

yes

---

> ### Author Response · Authors · 2021-03-17
> **Response to Reviewer 1**
>
> &nbsp;
>
> Thank you for understanding the core idea of this paper, and its merits, as well as providing the constructive comments that will help us improve the quality of the paper.
>
> &nbsp;
>
> * In Weakness: “A key weakness of the paper is that although it is primarily a methods paper, the method is only validated on a single, highly-specialised use case using a private dataset”.
>
> **Author Response #1**: Please refer to our response to Reviewer #4.
>
> &nbsp;
>
> * In detailed comments: " 'Criterion function" is highly unusual terminology - ' loss function" should be preferred.
>
> **Author Response #2**: We agree with you that criterion is a less popular expression. We used “criterion” function just to make it distinct from the overall loss function.
>
>
> &nbsp;
>
> * In detailed comments:  "The feature vectors drawn from where ?"
>
> **Author Response #3**: The feature of an image extracted usually refers to the feature vector before the last fc module. It will be interesting to investigate applying the regularization term on the intermediate features. However, please note that what flow inside a CNN when fed with an image are the features maps (instead of feature vectors); therefore, we may need extra layers to extract lower dimension feature vectors from the intermediate feature maps.
>
> &nbsp;
>
> * In detailed comments: “the batch size parameter? Will it affect the training?”
>
> **Author Response #4**: This is a very insightful and interesting observation! The batch size we used was 128. Apologies for omitting this in the paper.
>
> We agree with the reviewer’s opinion that small batch size will lead to smaller number of pairs of patches from the same slide that contribute to the regularization term in a batch, and consequently reduce the total number of such pairs in a training epoch. We hypothesize that when the batch size is large enough, the proposed regularization term can function well; in that case, there are sufficient pairs from the same slides being able to couple with each other in the regularization term.
>
> To further evaluate the impact of batch size, we conducted extra experiments using Resnet50 with color jitter, and the following table reports the AUC values, which presents something interesting. We could see for both methods, increasing the batch size indeed is beneficial for the performance. However, the AUC values of the baseline seem to saturate when the batch size is >32. On the other hand, when the batch size is small, the proposed SCR sees no significant superiority in performance in comparison to the baseline method; in contrast,, it is more significant when the batch size increases, since more patches from a slide are able to couple with each other in the regularization term. This trend empirically verifies the reviewer’s opinion.
>
> | Batch Size| 16| 32| 64| 128|
> | --- | --- | ---- | --- |--- |
> | Baseline | 0.908| 0.928 | 0.932 | 0.933 |
> | SCR | 0.911 |  0.923  | 0.943 | 0.953 |
>
> &nbsp;
>
> * In detailed comments: “Whether the regularization term affects the training stability”
>
> **Author Response #5**:  As shown in Table.3 in the paper, the SCR term will make the training performances converge to lower levels, but increase the test performance, i.e. implying it substantially increases the generalization ability of the models. We also provide the AUC values in the training and test sets over each training epoch, using ResNet50 with color jitter,  We observes that the curves are as smooth as those in Figure.2.
>
> | Epoch |   1 | 2 | 3 | 4 | 5 | 6 | 7 | 8 | 9 | 10 |
> | --- | --- | ---- | --- |--- | --- | --- | --- | --- | --- | --- |
> | Training | 0.883 |0.937 |0.958 | 0.975 | 0.979 | 0.982 | 0.984 | 0.987 | 0.986 | 0.988 |
> | Test     | 0.924 |	0.935 | 0.950 |	0.955	| 0.946 | 	0.957 | 	0.95 | 	0.955 | 	0.95 | 	0.952 |
>
> &nbsp;
>
> * In detailed comments: “It is rather confusing that the AUC scores in tables 1 and 2 do not seem to be highly correlated with the accuracy values. Is this because the resulting models are not well calibrated for some reason? Do the authors have any insight into this they can share?”
>
> **Author Response #6**: As stated in the paper, we uniformly use 0.5 as the threshold to determine the predictions to be positive or negative for all experiments, instead of using adaptive thresholds to get the optimal accuracy values. Also due to the distribution of the output probabilities, the AUC and the accuracy consequently are not always in accordance with each other. One extreme example is, all the predicted probabilities of ground-truth negative and positive samples are distributed between [0.1, 0.2] and [0.3, 0.4], respectively. They are perfectly separate, so the AUC value is 1; however, since they are all <0.5, the accuracy will only be 50%. That is why we state in the paper that AUC is a more informative performance metric.
>
> &nbsp;
>
> * “Do the authors intend to make an implementation of their regularisation term available?”
>
> **Author Response #7**: We will release the code.

---

> > ### Comment · AnonReviewer1 · 2021-03-22
> > **Response to authors**
> >
> > Thank you for providing some further experimental results and explanation. I especially appreciate the work on batch size experiments. There are a couple of things I would like to respond to:
> >
> > - I disagree with the authors that the word "feature" would be universally understood to mean the feature vector before the last FC layer. The word is very commonly used to refer to the outputs of any intermediate layer, so I certainly think it pays to be explicit here to avoid confusion.
> >
> > - I understand that the submitted paper is a proof of principle, but the use of a single highly specialised dataset certainly weakens the current manuscript
> >
> > - Regarding batch size, this is probably future work but I would suggest that a clearer experiment would be keeping the batch size the same (since as you have seen the batch size affects the performance of the baseline network) and altering its composition to increase or decrease the number of pairs from the same slide and see how this affects performance.

---

> > > ### Author Response · Authors · 2021-03-23
> > > **Response to Reviewer 1**
> > >
> > >
> > > Thank you very much for your suggestions.
> > >
> > > * We will investigate the impact of imposing the regularization term on the features of the intermediate layers as you suggested.
> > >
> > > * Regarding the suggestion about batch size, we observed from the extra experiments that when the batch size (>=32) was large enough, it had almost no impact on the baseline performance. However, your idea is interesting and possibly more rigorous, since by fixing the batch size and changing some elements with value ‘1' into ‘0' in the M matrix, we could directly deprive the impact of the batch size.

---

### Official Review · AnonReviewer2 · 2021-03-08

**Confidence:** 4
**Preliminary Rating:** 1
**Final Rating:** 2

**Summary:**

This paper formulates an additional loss term that can be used during the training of convolutional networks, to enforce the separation in the representation space between samples (patches) coming from the same input digital pathology image.
Performance is compared with baseline methods without any form of color augmentation and with a recent domain adaptation method.

**Strengths:**

* The method shows to improve performance when compared to using no color augmentation/standardization or using domain adaptation.
* The method is proposed to partly address the problem of dealing with small datasets.

**Weaknesses:**

* The use of English is sometimes poor (e.g., for automatically analysis..., each model was training for 10 epochs...)
* The problem addressed in this paper is essentially overfitting when a small set of slides is available, which can be partly mitigated with some data augmentation (not only color, but also elastic deformation, for example), or simply by using more slides, more than the 50 mentioned in the paper. I acknowledge that this might not be the case in some scenarios, but then if the focus of the paper is on dealing with small datasets, the method should also be compared with other approaches more specific for this problem, such as Bayesian methods.
* The experiment presented in section 2.2 (Figure 2) simply shows the basic concept of overfitting in medical imaging, when data from the same patient is present in both training and validation set.
* It is not clear how the performance is measured, for example, AUC is computed, but not clear if at patch level or at slide level, and how the predictions were aggregated if performance is reported at the slide level.
* Why are tiles sampled at 1024x1024 and then downscaled at 256x256? They could have been sampled directly at 10X magnification, rather than 40X.
* The proposed SCR method is presented as an alternative to traditional approaches like stain normalization or color augmentation. However, when SCR (results from Table 1) is compared with baseline using simple color jittering (results from Table 2), the baseline clearly outperforms the proposed method (for example, 0.834 vs. 0.890 for Resnet18, 0.889 vs. 0.899 for Resnet50, etc.).
* The paper does not provide any additional information, apart from a t-sne representation, about why the proposed loss term should work and what its effect is in practice.

**Deanonymize Review:**

no

**Final Rating Justification:**

I would like to thank the authors for clarifying some aspects of their paper.



**Justification Of The Preliminary Rating:**

The results and the explanation reported in this paper are not sufficient to convince about the actual effectiveness of the proposed method. The motivation for this work is not completely clear, and reported results do not fully confirm the superiority of the proposed method compared to basic color augmentation strategies.

**Paper Type:**

methodological development

**Special Issue:**

no

---

> ### Author Response · Authors · 2021-03-17
> **Response to Reviewer 2, Part 1 (1/2).**
>
> &nbsp;
>
> Thank you for reviewing the paper.
>
> &nbsp;
>
> * In Summary: “This paper formulates an additional loss term that can be used during the training of convolutional networks, to enforce the separation in the representation space between samples (patches) coming from the same input digital pathology image. "
>
> **Author Response #1**: We believe this summary does not accurately capture the core idea of the paper.  Please refer to the summaries from Reviewer 1 and Reviewer 4:
> in particular, “To enforce the separation in representation space” is not the essential purpose of the regularization term. The proposed regularization term essentially aims to drive the learnt features of patches away from the slide-specific feature spaces, which are not diagnostically relevant. Over-fitting to slide-specific feature is a major issue in whole slide image (WSI) analysis, since a large number patches are extracted from the same slides that share similar features.
>
> &nbsp;
>
> *  In Summary: “Performance is compared with baseline methods without any form of color augmentation and with a recent domain adaptation method. “
>
> **Author Response #2**: Apologies but this is incorrect: we compared performance with baseline methods with and without stain-normalization and color augmentation, and slide domain adversarial with and without color augmentation.
>
> Please note that the method we compared (Lafarge et al, SDA) is a domain-adversarial method, rather than “domain adaptation” as suggested. We believe that these are 2 distinct concepts: Domain adaption belongs to transfer learning, while domain-adversarial method as Lafarge et al.’s work shows is about adversarial training to make the features being agnostic about slides the patches come from.
>
> &nbsp;
>
> * In Strengths: “The method is proposed to partly address the problem of dealing with small datasets.
>
> **Author Response #3**: We are of the opinion that this comment  is not very accurate. Please refer to Author Response#1, the summaries from Reviewers 1 and 4. As Reviewer 4 suggested, the issue we deal with can be summarized as slide domain generalization. Further, the method is not intentionally proposed to deal with small datasets. Instead, what we try to convey in the paper’s Introduction is that common problems of WSI datasets are their huge image sizes and limited number of WSIs.
>
> &nbsp;
>
> * In Weakness: “The use of English is sometimes poor (e.g., for automatically analysis..., each model was training for 10 epochs...)“
>
> **Author Response #4**: Thank you. We will carefully proofread and correct our typographical errors and small English issues in the revision.
>
> &nbsp;
>
> * In Weakness: “The problem addressed in this paper is essentially overfitting when a small set of slides is available, which can be partly mitigated with some data augmentation (not only color, but also elastic deformation, for example), or simply by using more slides. I acknowledge that this might not be the case in some scenarios, but then if the focus of the paper is on dealing with small datasets, the method should also be compared with other approaches more specific for this problem, such as Bayesian methods.”
>
> **Author Response #5**: Again, we are of the opinion that the paper’s purpose and motivation are not precisely captured by the reviewer. Please refer to above Author Responses, and reviews from Reviewers 1 and 4. We do not intentionally deal with a small dataset (see Author Response#3). The patch similarities from the same slide are not only due to the colors, but also the cell morphologies, and even some less visible features  (e.g., noise in a certain range of frequency spectrum), and not all of these negative factors can be mitigated by the augmentation approaches suggested. In addition, as explained in the Introduction, since a slide is typically huge in size, a large number of patches will be cropped from the same slide, which share high correlation. In such cases, the number of slides will be comparably small from the perspective of machine learning.
>
> &nbsp;

---

> ### Author Response · Authors · 2021-03-17
> **Response to Reviewer 2, Part 2 (2/2).**
>
>
> &nbsp;
>
> * In Weakness: “The experiment presented in section 2.2 (Figure 2) simply shows the basic concept of overfitting in medical imaging, when data from the same patient is present in both training and validation set.”
>
> **Author Response #6**: These problems are exactly the common ones that the proposed strategy can be effective to address. The experiment presented in section 2.2 is simply an empirical demonstration to support the reasoning presented in the Introduction, i.e. that patches from same slide have high correlation. From the information perspective, the conditional entropy H(x|y) tends to be very small, when x, y are two patches from same slide. The shared features amongst the patches from the same slide that are not diagnostically relevant, are incorrectly learnt by the model as the important diagnostic features. That is why the validation performances in Fig.2(a) are almost the same with the training performance. However, the shared features hamper a trained model to be generalized to unseen  slides, and the proposed method aims to alleviate the negative impacts of these features.
>
> &nbsp;
>
> * In Weakness: “It is not clear how the performance is measured”.
>
> **Author Response #7**: We use the mean values of the predictions  of all patches in a slide as the slide prediction, and the corresponding slide-level performances are accordingly computed, as stated in section 3.1. Apologies, we should have made it more clear in section 3.1.
>
> &nbsp;
>
> * In Weakness: “Why are tiles sampled at 1024x1024 and then downscaled at 256x256?  Can directly use 10X“
>
> **Author Response #8**: Thank you. The downscaling from 1024x1024 to 256x256 generates the effect of 10x. We only saved the WSIs at 40X, so it was better to obtain 10X in this way.
>
> &nbsp;
>
> * In Weakness: “The proposed SCR method is presented as an alternative to traditional approaches like stain normalization or color augmentation. However, when SCR (results from Table 1) is compared with baseline using simple color jittering (results from Table 2), the baseline clearly outperforms the proposed method (for example, 0.834 vs. 0.890 for Resnet18, 0.889 vs. 0.899 for Resnet50, etc.). “
>
> **Author Response #9**: Stain normalization and color augmentation are pre-processing approaches, while the proposed method is implemented through the loss function. The former two are not mutually -exclusive to the proposed method, and they can be seen as complementary methods. After applying stain normalization or color augmentation, the issue of correlations between the patches from the same slide may still exist: since correlations may not only originate from color, but also from morphologies, that explains when combined with color jitter, the proposed method can further improve the performances. Further, for some applications, staining colors are important clues for diagnosis (e.g., Cytology images), in which applying color augmentation or stain normalization are not feasible since these operations will severely degrade the performance; however, the corresponding WSIs still have the issue of slide correlations due to similar morphologies inside an individual slide.
>
> &nbsp;
>
> * In Weakness: “The paper does not provide any additional information, apart from a t-sne representation, about why the proposed loss term should work and what its effect is in practice”.
>
> **Author Response #9**: We apologize for this lack of clarity and will seek to improve this in the revision. In the mean time, please refer to the Introduction as well as other parts in the paper, together with the comments from Reviewers 1 and 4, and the paper Lafarge et al.
>
> &nbsp;
>
> * “The motivation for this work is not completely clear”.
>
> **Author Response #9**: Thank you, we will try to improve the clarity in the revision. Please refer to the comments from Reviewers 1 and 4 as they have nicely summarized the novelty of our work.

---

### Meta-Review · Area_Chair1 · 2021-02-22

**Recommendation:** Accept (Poster)

**Metareview:**

The paper presents a scheme for training that aims to factor out correlations between CNN-derived feature vectors for patches in the same slide (same whole-slide image) within a dataset of histopathology images. The paper assumes that the patches will have higher correlations when then are taken from the same slide, which will lead to higher correlation between their extracted features. So, the paper penalizes the correlation between such features (Equation 3).

As some reviewers have pointed out, I'm unsure if the assumption underlying the proposed scheme really generalizes across all histopathology applications; it would depend on the variability of the textures within a slide. If a slide covers a large spatial region, the textures in distant regions within the image needn't be correlated. Also, patches obtained across two different slides could also be highly correlated.  As seen in Figure 3, all that the proposed method attempts to do is to reduce correlations between patches and have the patches spread over a larger region within the embedded space --- that seems to be the central idea underlying the regularization proposed in the paper and I'm unsure how useful it is going to be across all applications.

As some reviewers have suggested, more validation across datasets may be necessary.


**Paper Type:**

methodological development

---

### Decision · Program_Chairs · 2021-03-31

Accept